# On Decomposing the Proximal Map

**Yaoliang Yu**
Department of Computing Science, University of Alberta, Edmonton AB T6G 2E8, Canada
`yaoliang@cs.ualberta.ca`

## Abstract

The proximal map is the key step in gradient-type algorithms, which have become prevalent in large-scale high-dimensional problems. For simple functions this proximal map is available in closed-form while for more complicated functions it can become highly nontrivial. Motivated by the need of combining regularizers to simultaneously induce different types of structures, this paper initiates a systematic investigation of when the proximal map of a sum of functions decomposes into the composition of the proximal maps of the individual summands. We not only unify a few known results scattered in the literature but also discover several new decompositions obtained almost effortlessly from our theory.

## 1 Introduction

Regularization has become an indispensable part of modern machine learning algorithms. For example, the $\ell_2$-regularizer for kernel methods [1] and the $\ell_1$-regularizer for sparse methods [2] have led to immense successes in various fields. As real data become more and more complex, different types of regularizers, usually nonsmooth functions, have been designed. In many applications, it is thus desirable to combine regularizers, usually taking their sum, to promote different structures simultaneously.

Since many interesting regularizers are nonsmooth, they are harder to optimize numerically, especially in large-scale high-dimensional settings. Thanks to recent advances [3–5], gradient-type algorithms have been generalized to take nonsmooth regularizers explicitly into account. And due to their cheap per-iteration cost (usually linear-time), these algorithms have become prevalent in many fields recently. The key step of such gradient-type algorithms is to compute the proximal map (of the nonsmooth regularizer), which is available in closed-form for some specific regularizers. However, the proximal map becomes highly nontrivial when we start to combine regularizers.

The main goal of this paper is to systematically investigate when the proximal map of a sum of functions decomposes into the composition of the proximal maps of the individual functions, which we simply term prox-decomposition. Our motivation comes from a few known decomposition results scattered in the literature [6–8], all in the form of our interest. The study of such prox-decompositions is not only of mathematical interest, but also the backbone of popular gradient-type algorithms [3–5]. More importantly, a precise understanding of this decomposition will shed light on how we should combine regularizers, taking computational efforts explicitly into account.

After setting the context in Section 2, we motivate the decomposition rule with some justifications, as well as some cautionary results. Based on a sufficient condition presented in Section 3.1, we study how "invariance" of the subdifferential of one function would lead to nontrivial prox-decompositions. Specifically, we prove in Section 3.3 that when the subdifferential of one function is scaling invariant, then the prox-decomposition always holds if and only if another function is radial—which is, quite unexpectedly, exactly the same condition proven recently for the validity of the representer theorem in the context of kernel methods [9, 10]. The generalization to cone invariance is considered in Section 3.4, and enables us to recover most known prox-decompositions, as well as some new ones falling out quite naturally.

Our notations are mostly standard. We use $\iota_C(x)$ for the indicator function that takes 0 if $x \in C$ and $\infty$ otherwise, and $\mathbf{1}_C(x)$ for the indicator that takes 1 if $x \in C$ and 0 otherwise. The symbol Id stands for the identity map and the extended real line $\mathbb{R} \cup \{\infty\}$ is denoted as $\bar{\mathbb{R}}$. Throughout the paper we denote $\partial f(\mathbf{x})$ as the subdifferential of the function $f$ at point $\mathbf{x}$.

## 2 Preliminary

Let our domain be some (real) Hilbert space $(\mathcal{H}, \langle \cdot, \cdot \rangle)$, with the induced Hilbertian norm $\| \cdot \|$. If needed, we will assume some fixed orthonormal basis $\{\mathbf{e}_i\}_{i \in I}$ is chosen for $\mathcal{H}$, so that for $\mathbf{x} \in \mathcal{H}$ we are able to refer to its "coordinates" $x_i = \langle \mathbf{x}, \mathbf{e}_i \rangle$.

For any closed convex proper function $f : \mathcal{H} \to \bar{\mathbb{R}}$, we define its Moreau envelop as [11]

$$\forall \mathbf{y} \in \mathcal{H}, \ \ \mathsf{M}_f(\mathbf{y}) \ = \ \min_{\mathbf{x} \in \mathcal{H}} \tfrac{1}{2}\|\mathbf{x} - \mathbf{y}\|^2 + f(\mathbf{x}), \tag{1}$$

and the related proximal map

$$\mathsf{P}_f(\mathbf{y}) \ = \ \operatorname*{argmin}_{\mathbf{x} \in \mathcal{H}} \tfrac{1}{2}\|\mathbf{x} - \mathbf{y}\|^2 + f(\mathbf{x}). \tag{2}$$

Due to the strong convexity of $\| \cdot \|^2$ and the closedness and convexity of $f$, $\mathsf{P}_f(\mathbf{y})$ always exists and is unique. Note that $\mathsf{M}_f : \mathcal{H} \to \mathbb{R}$ while $\mathsf{P}_f : \mathcal{H} \to \mathcal{H}$. When $f = \iota_C$ is the indicator of some closed convex set $C$, the proximal map reduces to the usual projection. Perhaps the most interesting property of $\mathsf{M}_f$, known as Moreau's identity, is the following decomposition [11]

$$\mathsf{M}_f(\mathbf{y}) + \mathsf{M}_{f^*}(\mathbf{y}) = \tfrac{1}{2}\|\mathbf{y}\|^2, \tag{3}$$

where $f^*(\mathbf{z}) = \sup_{\mathbf{x}} \langle \mathbf{x}, \mathbf{z} \rangle - f(\mathbf{x})$ is the Fenchel conjugate of $f$. It can be shown that $\mathsf{M}_f$ is Frechét differentiable, hence taking derivative *w.r.t.* $\mathbf{y}$ in both sides of (3) yields

$$\mathsf{P}_f(\mathbf{y}) + \mathsf{P}_{f^*}(\mathbf{y}) = \mathbf{y}. \tag{4}$$

## 3 Main Results

Our main goal is to investigate and understand the equality (we always assume $f + g \not\equiv \infty$)

$$\mathsf{P}_{f+g} \overset{?}{=} \mathsf{P}_f \circ \mathsf{P}_g \overset{?}{=} \mathsf{P}_g \circ \mathsf{P}_f, \tag{5}$$

where $f, g \in \Gamma_0$, the set of all closed convex proper functions on $\mathcal{H}$, and $f \circ g$ denotes the mapping composition. We present first some cautionary results.

Note that $\mathsf{P}_f = (\mathsf{Id} + \partial f)^{-1}$, hence under minor technical assumptions $\mathsf{P}_{f+g} = (\mathsf{P}_{2f}^{-1} + \mathsf{P}_{2g}^{-1})^{-1} \circ 2\mathsf{Id}$. However, computationally this formula is of little use. On the other hand, it is possible to develop forward-backward splitting procedures[1] to numerically compute $\mathsf{P}_{f+g}$, using only $\mathsf{P}_f$ and $\mathsf{P}_g$ as subroutines [12]. Our focus is on the exact closed-form formula (5). Interestingly, under some "shrinkage" assumption, the prox-decomposition (5), even if not necessarily hold, can still be used in subgradient algorithms [13].

Our first result is encouraging:

**Proposition 1.** *If $\mathcal{H} = \mathbb{R}$, then for any $f, g \in \Gamma_0$, there exists $h \in \Gamma_0$ such that $\mathsf{P}_h = \mathsf{P}_f \circ \mathsf{P}_g$.*

*Proof:* In fact, Moreau [11, Corollary 10.c] proved that $\mathsf{P} : \mathcal{H} \to \mathcal{H}$ is a proximal map iff it is nonexpansive and it is the subdifferential of some convex function in $\Gamma_0$. Although the latter condition in general is not easy to verify, it reduces to monotonic increasing when $\mathcal{H} = \mathbb{R}$ (note that $\mathsf{P}$ must be continuous). Since both $\mathsf{P}_f$ and $\mathsf{P}_g$ are increasing and nonexpansive, it follows easily that so is $\mathsf{P}_f \circ \mathsf{P}_g$, hence the existence of $h \in \Gamma_0$ so that $\mathsf{P}_h = \mathsf{P}_f \circ \mathsf{P}_g$. ∎

In a general Hilbert space $\mathcal{H}$, we again easily conclude that the composition $\mathsf{P}_f \circ \mathsf{P}_g$ is always a nonexpansion, which means that it is "close" to be a proximal map. This justifies the composition $\mathsf{P}_f \circ \mathsf{P}_g$ as a candidate for the decomposition of $\mathsf{P}_{f+g}$. However, we note that Proposition 1 indeed can fail already in $\mathbb{R}^2$:

**Example 1.** *Let $\mathcal{H} = \mathbb{R}^2$. Let $f = \iota_{\{x_1 = x_2\}}$ and $g = \iota_{\{x_2 = 0\}}$. Clearly both $f$ and $g$ are in $\Gamma_0$. The proximal maps in this case are simply projections: $\mathsf{P}_f(\mathbf{x}) = (\frac{x_1 + x_2}{2}, \frac{x_1 + x_2}{2})$ and $\mathsf{P}_g(\mathbf{x}) = (x_1, 0)$. Therefore $\mathsf{P}_f(\mathsf{P}_g(\mathbf{x})) = (\frac{x_1}{2}, \frac{x_1}{2})$. We easily verify that the inequality*

$$\|\mathsf{P}_f(\mathsf{P}_g(\mathbf{x})) - \mathsf{P}_f(\mathsf{P}_g(\mathbf{y}))\|^2 \leq \langle \mathsf{P}_f(\mathsf{P}_g(\mathbf{x})) - \mathsf{P}_f(\mathsf{P}_g(\mathbf{y})), \mathbf{x} - \mathbf{y} \rangle$$

*is* not *always true, contradiction if $\mathsf{P}_f \circ \mathsf{P}_g$ was a proximal map [11, Eq. (5.3)].*

Even worse, when Proposition 1 does hold, in general we can *not* expect the decomposition (5) to be true without additional assumptions.

**Example 2.** *Let $\mathcal{H} = \mathbb{R}$ and $\mathsf{q}(x) = \frac{1}{2}x^2$. It is easily seen that $\mathsf{P}_{\lambda\mathsf{q}}(x) = \frac{1}{1+\lambda}x$. Therefore $\mathsf{P}_\mathsf{q} \circ \mathsf{P}_\mathsf{q} = \frac{1}{4}\mathsf{Id} \neq \frac{1}{3}\mathsf{Id} = \mathsf{P}_{\mathsf{q}+\mathsf{q}}$. We will give an explanation for this failure of composition shortly.*

Nevertheless, as we will see, the equality in (5) does hold in many scenarios, and an interesting theory can be suitably developed.

### 3.1 A Sufficient Condition

We start with a sufficient condition that yields (5). This result, although easy to obtain, will play a key role in our subsequent development.

Using the first order optimality condition and the definition of the proximal map (2), we have

$$\mathsf{P}_{f+g}(\mathbf{y}) - \mathbf{y} + \partial(f + g)(\mathsf{P}_{f+g}(\mathbf{y})) \ni 0 \tag{6}$$

$$\mathsf{P}_g(\mathbf{y}) - \mathbf{y} + \partial g(\mathsf{P}_g(\mathbf{y})) \ni 0 \tag{7}$$

$$\mathsf{P}_f(\mathsf{P}_g(\mathbf{y})) - \mathsf{P}_g(\mathbf{y}) + \partial f(\mathsf{P}_f(\mathsf{P}_g(\mathbf{y}))) \ni 0. \tag{8}$$

Adding the last two equations we obtain

$$\mathsf{P}_f(\mathsf{P}_g(\mathbf{y})) - \mathbf{y} + \partial g(\mathsf{P}_g(\mathbf{y})) + \partial f(\mathsf{P}_f(\mathsf{P}_g(\mathbf{y}))) \ni 0. \tag{9}$$

Comparing (6) and (9) gives us

**Theorem 1.** *A sufficient condition for $\mathsf{P}_{f+g} = \mathsf{P}_f \circ \mathsf{P}_g$ is*

$$\forall\, \mathbf{x} \in \mathcal{H},\ \partial g(\mathsf{P}_f(\mathbf{x})) \supseteq \partial g(\mathbf{x}). \tag{10}$$

*Proof:* Let $\mathbf{x} = \mathsf{P}_g(\mathbf{y})$. Then by (9) and the subdifferential rule $\partial(f + g) \supseteq \partial f + \partial g$ we verify that $\mathsf{P}_f(\mathsf{P}_g(\mathbf{y}))$ satisfies (6), hence follows $\mathsf{P}_{f+g} = \mathsf{P}_f \circ \mathsf{P}_g$ since the proximal map is single-valued. ∎

We note that a special form of our sufficient condition has appeared in the proof of [8, Theorem 1], whose main result also follows immediately from our Theorem 4 below. Let us fix $f$, and define

$$\mathcal{K}_f = \{g \in \Gamma_0 : f + g \not\equiv \infty, (f, g) \text{ satisfy } (10)\}.$$

Immediately we have

**Proposition 2.** *For any $f \in \Gamma_0$, $\mathcal{K}_f$ is a cone. Moreover, if $g_1 \in \mathcal{K}_f, g_2 \in \mathcal{K}_f$, $f + g_1 + g_2 \not\equiv \infty$ and $\partial(g_1 + g_2) = \partial g_1 + \partial g_2$, then $g_1 + g_2 \in \mathcal{K}_f$ too.*

The condition $\partial(g_1 + g_2) = \partial g_1 + \partial g_2$ in Proposition 2 is purely technical; it is satisfied when, say $g_1$ is continuous at a single, arbitrary point in $\text{dom } g_1 \cap \text{dom } g_2$. For comparison purpose, we note that it is not clear how $\mathsf{P}_{f+g+h} = \mathsf{P}_f \circ \mathsf{P}_{g+h}$ would follow from $\mathsf{P}_{f+g} = \mathsf{P}_f \circ \mathsf{P}_g$ and $\mathsf{P}_{f+h} = \mathsf{P}_f \circ \mathsf{P}_h$. This is the main motivation to consider the sufficient condition (10). In particular

**Definition 1.** *We call $f \in \Gamma_0$ self-prox-decomposable (s.p.d.) if $f \in \mathcal{K}_{\alpha f}$ for all $\alpha > 0$.*

For any s.p.d. $f$, since $\mathcal{K}_f$ is a cone, $\beta f \in \mathcal{K}_{\alpha f}$ for all $\alpha, \beta \geq 0$. Consequently, $\mathsf{P}_{(\alpha+\beta)f} = \mathsf{P}_{\beta f} \circ \mathsf{P}_{\alpha f} = \mathsf{P}_{\alpha f} \circ \mathsf{P}_{\beta f}$.

**Remark 1.** *A weaker definition for s.p.d. is to require $f \in \mathcal{K}_f$, from which we conclude that $\beta f \in \mathcal{K}_f$ for all $\beta \geq 0$, in particular $\mathsf{P}_{(m+n)f} = \mathsf{P}_{nf} \circ \mathsf{P}_{mf} = \mathsf{P}_{mf} \circ \mathsf{P}_{nf}$ for all natural numbers $m$ and $n$. The two definitions coincide for positive homogeneous functions. We have not been able to construct a function that satisfies this weaker definition but not the stronger one in Definition 1.*

**Example 3.** *We easily verify that all affine functions $\ell = \langle \cdot, \mathbf{a} \rangle + b$ are s.p.d., in fact, they are the only differentiable functions that are s.p.d., which explains why Example 2 must fail. Another trivial class of s.p.d. functions are projectors to closed convex sets. Also, univariate gauges[2] are s.p.d., due to Theorem 4 below. Some multivariate s.p.d. functions are given in Remark 5 below.*

The next example shows that (10) is not necessary.

**Example 4.** *Fix* $\mathbf{z} \in \mathcal{H}$, $f = \iota_{\{\mathbf{z}\}}$, *and* $g \in \Gamma_0$ *with full domain. Clearly for any* $\mathbf{x} \in \mathcal{H}$, $\mathsf{P}_{f+g}(\mathbf{x}) = \mathbf{z} = \mathsf{P}_f[\mathsf{P}_g(\mathbf{x})]$. *However, since* $\mathbf{x}$ *is arbitrary,* $\partial g(\mathsf{P}_f(\mathbf{x})) = \partial g(\mathbf{z}) \not\supseteq \partial g(\mathbf{x})$ *if* $g$ *is not linear.*

On the other hand, if $f, g$ are differentiable, then we actually have equality in (10), which is clearly necessary in this case. Since convex functions are almost everywhere differentiable (in the interior of their domain), we expect the sufficient condition (10) to be necessary "almost everywhere" too.

Thus we see that the key for the decomposition (5) to hold is to let the proximal map of $f$ and the subdifferential of $g$ "interact well" in the sense of (10). Interestingly, both are fully equivalent to the function itself.

**Proposition 3** ([11, §8]). *Let* $f, g \in \Gamma_0$. $f = g + c$ *for some* $c \in \mathbb{R} \iff \partial f \subseteq \partial g \iff \mathsf{P}_f = \mathsf{P}_g$.

*Proof:* The first implication is clear. The second follows from the optimality condition $\mathsf{P}_f = (\mathsf{Id} + \partial f)^{-1}$. Lastly, $\mathsf{P}_f = \mathsf{P}_g$ implies that $\mathsf{M}_{f^*} = \mathsf{M}_{g^*} - c$ for some $c \in \mathbb{R}$ (by integration). Conjugating we get $f = g + c$ for some $c \in \mathbb{R}$. ■

Therefore some properties of the proximal map will transfer to some properties of the function $f$ itself, and vice versa. The next result is easy to obtain, and appeared essentially in [14].

**Proposition 4.** *Let* $f \in \Gamma_0$ *and* $\mathbf{x} \in \mathcal{H}$ *be arbitrary, then*

i). $\mathsf{P}_f$ *is odd iff* $f$ *is even;*

ii). $\mathsf{P}_f(U\mathbf{x}) = U\mathsf{P}_f(\mathbf{x})$ *for all unitary* $U$ *iff* $f(U\mathbf{x}) = f(\mathbf{x})$ *for all unitary* $U$;

iii). $\mathsf{P}_f(Q\mathbf{x}) = Q\mathsf{P}_f(\mathbf{x})$ *for all permutation* $Q$ *(under some fixed basis) iff* $f$ *is permutation invariant, that is* $f(Q\mathbf{x}) = f(\mathbf{x})$ *for all permutation* $Q$.

In the following, we will put some invariance assumptions on the subdifferential of $g$ and accordingly find the right family of $f$ whose proximal map "respects" that invariance. This way we will meet (10) by construction therefore effortlessly have the decomposition (5).

## 3.2 No Invariance

To begin with, consider first the trivial case where no invariance on the subdifferential of $g$ is assumed. This is equivalent as requiring (10) to hold for all $g \in \Gamma_0$. Not surprisingly, we end up with a trivial choice of $f$.

**Theorem 2.** *Fix* $f \in \Gamma_0$. $\mathsf{P}_{f+g} = \mathsf{P}_f \circ \mathsf{P}_g$ *for all* $g \in \Gamma_0$ *if and only if*

- $\dim(\mathcal{H}) \geq 2$; $f \equiv c$ *or* $f = \iota_{\{\mathbf{w}\}} + c$ *for some* $c \in \mathbb{R}$ *and* $\mathbf{w} \in \mathcal{H}$;

- $\dim(\mathcal{H}) = 1$ *and* $f = \iota_C + c$ *for some closed and convex set* $C$ *and* $c \in \mathbb{R}$.

*Proof:* $\Leftarrow$: Straightforward calculations, see [15] for details.

$\Rightarrow$: We first prove that $f$ is constant on its domain even when $g$ is restricted to indicators. Indeed, let $\mathbf{x} \in \mathrm{dom}\, f$ and take $g = \iota_{\{\mathbf{x}\}}$. Then $\mathbf{x} = \mathsf{P}_{f+g}(\mathbf{x}) = \mathsf{P}_f[\mathsf{P}_g(\mathbf{x})] = \mathsf{P}_f(\mathbf{x})$, meaning that $\mathbf{x} \in \arg\min f$. Since $\mathbf{x} \in \mathrm{dom}\, f$ is arbitrary, $f$ is constant on its domain. The case $\dim(\mathcal{H}) = 1$ is complete. We consider the other case where $\dim(\mathcal{H}) \geq 2$ and $\mathrm{dom}\, f$ contains at least two points. If $\mathrm{dom}\, f \neq \mathcal{H}$, there exists $\mathbf{z} \notin \mathrm{dom}\, f$ such that $\mathsf{P}_f(\mathbf{z}) = \mathbf{y}$ for some $\mathbf{y} \in \mathrm{dom}\, f$, and closed convex set $C \cap \mathrm{dom}\, f \neq \emptyset$ with $\mathbf{y} \notin C \ni \mathbf{z}$. Let $g = \iota_C$ we obtain $\mathsf{P}_{f+g}(\mathbf{z}) \in C \cap \mathrm{dom}\, f$ while $\mathsf{P}_f(\mathsf{P}_g(\mathbf{z})) = \mathsf{P}_f(\mathbf{z}) = \mathbf{y} \notin C$, contradiction. ■

Observe that the decomposition (5) is not symmetric in $f$ and $g$, also reflected in the next result:

**Theorem 3.** *Fix* $g \in \Gamma_0$. $\mathsf{P}_{f+g} = \mathsf{P}_f \circ \mathsf{P}_g$ *for all* $f \in \Gamma_0$ *iff* $g$ *is a continuous affine function.*

*Proof:* $\Rightarrow$: If $g = \langle \cdot, \mathbf{a} \rangle + c$, then $\mathsf{P}_g(\mathbf{x}) = \mathbf{x} - \mathbf{a}$. Easy calculation reveals that $\mathsf{P}_{f+g}(\mathbf{x}) = \mathsf{P}_f(\mathbf{x} - \mathbf{a}) = \mathsf{P}_f[\mathsf{P}_g(\mathbf{x})]$.

$\Leftarrow$: The converse is true even when $f$ is restricted to continuous linear functions. Indeed, let $\mathbf{a} \in \mathcal{H}$ be arbitrary and consider $f = \langle \cdot, \mathbf{a} \rangle$. Then $\mathsf{P}_{f+g}(\mathbf{x}) = \mathsf{P}_g(\mathbf{x} - \mathbf{a}) = \mathsf{P}_f(\mathsf{P}_g(\mathbf{x})) = \mathsf{P}_g(\mathbf{x}) - \mathbf{a}$. Letting $\mathbf{a} = \mathbf{x}$ yields $\mathsf{P}_g(\mathbf{x}) = \mathbf{x} + \mathsf{P}_g(\mathbf{0}) = \mathsf{P}_{\langle \cdot, -\mathsf{P}_g(\mathbf{0}) \rangle}(\mathbf{x})$. Therefore by Proposition 3 we know $g$ is equal to a continuous affine function. ■

Naturally, the next step is to put invariance assumptions on the subdifferential of $g$, effectively restricting the function class of $g$. As a trade off, the function class of $f$, that satisfies (10), becomes larger so that nontrivial results will arise.

### 3.3 Scaling Invariance

The first invariance property we consider is scaling-invariance. What kind of convex functions have their subdifferential invariant to (positive) scaling? Assuming $\mathbf{0} \in \operatorname{dom} g$ and by simple integration

$$g(t\mathbf{x}) - g(\mathbf{0}) = \int_0^t g'(s\mathbf{x})\mathrm{d}s = \int_0^t \langle \partial g(s\mathbf{x}), \mathbf{x} \rangle \, \mathrm{d}s = t \cdot [g(\mathbf{x}) - g(\mathbf{0})],$$

where the last equality follows from the scaling invariance of the subdifferential of $g$. Therefore, up to some additive constant, $g$ is positive homogeneous (p.h.). On the other hand, if $g \in \Gamma_0$ is p.h. (automatically $\mathbf{0} \in \operatorname{dom} g$), then from definition we verify that $\partial g$ is scaling-invariant. Therefore, under the scaling-invariance assumption, the right function class for $g$ is the set of all p.h. functions in $\Gamma_0$, up to some additive constant. Consequently, the right function class for $f$ is to have the proximal map $\mathsf{P}_f(\mathbf{x}) = \lambda \cdot \mathbf{x}$ for some $\lambda \in [0,1]$ that may depend on $\mathbf{x}$ as well[3]. The next theorem completely characterizes such functions.

**Theorem 4.** *Let $f \in \Gamma_0$. Consider the statements*

i). *$f = h(\| \cdot \|)$ for some increasing function $h : \mathbb{R}_+ \to \bar{\mathbb{R}}$;*

ii). *$\mathbf{x} \perp \mathbf{y} \implies f(\mathbf{x} + \mathbf{y}) \geq f(\mathbf{y})$;*

iii). *$\mathsf{P}_f(\mathbf{u}) = \lambda \cdot \mathbf{u}$ for some $\lambda \in [0,1]$ (that may itself depend on $\mathbf{u}$);*

iv). *$\mathbf{0} \in \operatorname{dom} f$ and $\mathsf{P}_{f+\kappa} = \mathsf{P}_f \circ \mathsf{P}_\kappa$ for all p.h. (up to some additive constant) function $\kappa \in \Gamma_0$.*

*Then we have i) $\implies$ ii) $\iff$ iii) $\iff$ iv). Moreover, when $\dim(\mathcal{H}) \geq 2$, ii) $\implies$ i) as well, in which case $\mathsf{P}_f(\mathbf{u}) = \mathsf{P}_h(\|\mathbf{u}\|)/\|\mathbf{u}\| \cdot \mathbf{u}$ (where we interpret $0/0 = 0$).*

**Remark 2.** *When $\dim(\mathcal{H}) = 1$, ii) is equivalent as requiring $f$ to attain its minimum at 0, in which case the implication ii) $\implies$ iv), under the redundant condition that $f$ is differentiable, was proved by Combettes and Pesquet [14, Proposition 3.6]. The implication ii) $\implies$ iii) also generalizes [14, Corollary 2.5], where only the case $\dim(\mathcal{H}) = 1$ and $f$ differentiable is considered. Note that there exists non-even $f$ that satisfies Theorem 4 when $\dim(\mathcal{H}) = 1$. Such is impossible for $\dim(\mathcal{H}) \geq 2$, in which case any $f$ that satisfies Theorem 4 must also enjoy all properties listed in Proposition 4.*

*Proof:* i) $\implies$ ii): $\mathbf{x} \perp \mathbf{y} \implies \|\mathbf{x} + \mathbf{y}\| \geq \|\mathbf{y}\|$.

ii) $\implies$ iii): Indeed, by definition

$$\mathsf{M}_f(\mathbf{u}) = \min_{\mathbf{x}} \tfrac{1}{2}\|\mathbf{x} - \mathbf{u}\|^2 + f(\mathbf{x}) = \min_{\mathbf{u}^\perp, \lambda} \tfrac{1}{2}\|\mathbf{u}^\perp + \lambda\mathbf{u} - \mathbf{u}\|^2 + f(\mathbf{u}^\perp + \lambda\mathbf{u})$$

$$= \min_\lambda \tfrac{1}{2}\|\lambda\mathbf{u} - \mathbf{u}\|^2 + f(\lambda\mathbf{u}) = \min_{\lambda \in [0,1]} \tfrac{1}{2}(\lambda - 1)^2\|\mathbf{u}\|^2 + f(\lambda\mathbf{u}),$$

where the third equality is due to ii), and the nonnegative constraint in the last equality can be seen as follows: For any $\lambda < 0$, by increasing it to 0 we can only decrease both terms; similar argument for $\lambda > 1$. Therefore there exists $\lambda \in [0,1]$ such that $\lambda\mathbf{u}$ minimizes the Moreau envelop $\mathsf{M}_f$ hence we have $\mathsf{P}_f(\mathbf{u}) = \lambda\mathbf{u}$ due to uniqueness.

iii) $\implies$ iv): Note first that iii) implies $\mathbf{0} \in \partial f(\mathbf{0})$, therefore $\mathbf{0} \in \operatorname{dom} f$. Since the subdifferential of $\kappa$ is scaling-invariant, iii) implies the sufficient condition (10) hence iv).

iv) $\implies$ iii): Fix $\mathbf{y}$ and construct the gauge function

$$\kappa(\mathbf{z}) = \begin{cases} 0, & \text{if } \mathbf{z} = \lambda \cdot \mathbf{y} \text{ for some } \lambda \geq 0 \\ \infty, & \text{otherwise} \end{cases}.$$

Then $\mathsf{P}_\kappa(\mathbf{y}) = \mathbf{y}$, hence $\mathsf{P}_f(\mathsf{P}_\kappa(\mathbf{y})) = \mathsf{P}_f(\mathbf{y}) = \mathsf{P}_{f+\kappa}(\mathbf{y})$ by iv). On the other hand,

$$\mathsf{M}_{f+\kappa}(\mathbf{y}) = \min_{\mathbf{x}} \tfrac{1}{2}\|\mathbf{x} - \mathbf{y}\|_2^2 + f(\mathbf{x}) + \kappa(\mathbf{x}) = \min_{\lambda \geq 0} \tfrac{1}{2}\|\lambda\mathbf{y} - \mathbf{y}\|_2^2 + f(\lambda\mathbf{y}). \qquad (11)$$

Take $\mathbf{y} = \mathbf{0}$ we obtain $\mathsf{P}_{f+\kappa}(\mathbf{0}) = \mathbf{0}$. Thus $\mathsf{P}_f(\mathbf{0}) = \mathbf{0}$, *i.e.* $\mathbf{0} \in \partial f(\mathbf{0})$, from which we deduce that $\mathsf{P}_f(\mathbf{y}) = \mathsf{P}_{f+\kappa}(\mathbf{y}) = \lambda\mathbf{y}$ for some $\lambda \in [0, 1]$, since $f(\lambda\mathbf{y})$ in (11) is increasing on $[1, \infty[$.

iii) $\implies$ ii): First note that iii) implies that $\mathsf{P}_f(\mathbf{0}) = \mathbf{0}$ hence $\mathbf{0} \in \partial f(\mathbf{0})$, in particular, $\mathbf{0} \in \operatorname{dom} f$. If $\dim(\mathcal{H}) = 1$ we are done, so we assume $\dim(\mathcal{H}) \geq 2$ in the rest of the proof. In this case, it is known, *cf.* [9, Theorem 1] or [10, Theorem 3], that ii) $\iff$ i) (even without assuming $f$ convex). All we left is to prove iii) $\implies$ ii) or equivalently i), for the case $\dim(\mathcal{H}) \geq 2$.

We first prove the case when $\operatorname{dom} f = \mathcal{H}$. By iii), $\mathsf{P}_f(\mathbf{x}) = \lambda\mathbf{x}$ for some $\lambda \in [0, 1]$ (which may depend on $\mathbf{x}$ as well). Using the first order optimality condition for the proximal map we have $0 \in \lambda\mathbf{x} - \mathbf{x} + \partial f(\lambda\mathbf{x})$, that is $(\frac{1}{\lambda} - 1)\mathbf{y} \in \partial f(\mathbf{y})$ for each $\mathbf{y} \in \operatorname{ran}(\mathsf{P}_f) = \mathcal{H}$ due to our assumption $\operatorname{dom} f = \mathcal{H}$. Now for any $\mathbf{x} \perp \mathbf{y}$, by the definition of the subdifferential,

$$f(\mathbf{x} + \mathbf{y}) \geq f(\mathbf{y}) + \langle \mathbf{x}, \partial f(\mathbf{y}) \rangle = f(\mathbf{y}) + \langle \mathbf{x}, (\tfrac{1}{\lambda} - 1)\mathbf{y} \rangle = f(\mathbf{y}).$$

For the case when $\operatorname{dom} f \subset \mathcal{H}$, we consider the proximal average [16]

$$g = \mathsf{A}(f, \mathsf{q}) = [(\tfrac{1}{2}(f^* + \mathsf{q})^* + \tfrac{1}{4}\mathsf{q})^* - \mathsf{q}]^*, \tag{12}$$

where $\mathsf{q} = \frac{1}{2}\|\cdot\|^2$. Importantly, since $\mathsf{q}$ is defined on the whole space, the proximal average $g$ has full domain too [16, Corollary 4.7]. Moreover, $\mathsf{P}_g(\mathbf{x}) = \frac{1}{2}\mathsf{P}_f(\mathbf{x}) + \frac{1}{4}\mathbf{x} = (\frac{1}{2}\lambda + \frac{1}{4})\mathbf{x}$. Therefore by our previous argument, $g$ satisfies ii) hence also i). It is easy to check that i) is preserved under taking the Fenchel conjugation (note that the convexity of $f$ implies that of $h$). Since we have shown that $g$ satisfies i), it follows from (12) that $f$ satisfies i) hence also ii).

As mentioned, when $\dim(\mathcal{H}) \geq 2$, the implication ii) $\implies$ i) was shown in [9, Theorem 1]. The formula $\mathsf{P}_f(\mathbf{u}) = \mathsf{P}_h(\|\mathbf{u}\|)/\|\mathbf{u}\| \cdot \mathbf{u}$ for $f = h(\|\cdot\|)$ follows from straightforward calculation. ∎

We now discuss some applications of Theorem 4. When $\dim(\mathcal{H}) \geq 2$, iii) in Theorem 4 automatically implies that the scalar constant $\lambda$ depends on $\mathbf{x}$ only through its norm. This fact, although not entirely obvious, does have a clear geometric picture:

**Corollary 1.** *Let $\dim(\mathcal{H}) \geq 2$, $C \subseteq \mathcal{H}$ be a closed convex set that contains the origin. Then the projection onto $C$ is simply a shrinkage towards the origin iff $C$ is a ball (of the norm $\|\cdot\|$).*

*Proof:* Let $f = \iota_C$ and apply Theorem 4. ∎

**Example 5.** *As usual, denote $\mathsf{q} = \frac{1}{2}\|\cdot\|^2$. In many applications, in addition to the regularizer $\kappa$ (usually a gauge), one adds the $\ell_2^2$ regularizer $\lambda\mathsf{q}$ either for stability or grouping effect or strong convexity. This incurs no computational cost in the sense of computing the proximal map: We easily compute that $\mathsf{P}_{\lambda\mathsf{q}} = \frac{1}{\lambda+1}\mathsf{Id}$. By Theorem 4, for any gauge $\kappa$, $\mathsf{P}_{\kappa+\lambda\mathsf{q}} = \frac{1}{\lambda+1}\mathsf{P}_\kappa$, whence it is also clear that adding an extra $\ell_2$ regularizer tends to double "shrink" the solution. In particular, let $\mathcal{H} = \mathbb{R}^d$ and take $\kappa = \|\cdot\|_1$ (the sum of absolute values) we recover the proximal map for the elastic-net regularizer [17].*

**Example 6.** *The Berhu regularizer*

$$h(x) = |x|\mathbf{1}_{|x|<\gamma} + \frac{x^2+\gamma^2}{2\gamma}\mathbf{1}_{|x|\geq\gamma} = |x| + \frac{(|x|-\gamma)^2}{2\gamma}\mathbf{1}_{|x|\geq\gamma},$$

*being the reverse of Huber's function, is proposed in [18] as a bridge between the lasso ($\ell_1$ regularization) and ridge regression ($\ell_2^2$ regularization). Let $f(x) = h(x) - |x|$. Clearly, $f$ satisfies ii) of Theorem 4 (but not differentiable), hence*

$$\mathsf{P}_h = \mathsf{P}_f \circ \mathsf{P}_{|\cdot|},$$

*whereas simple calculation verifies that*

$$\mathsf{P}_f(x) = \operatorname{sign}(x) \cdot \min\{|x|, \tfrac{\gamma}{1+\gamma}(|x|+1)\},$$

*and of course $\mathsf{P}_{|\cdot|}(x) = \operatorname{sign}(x) \cdot \max\{|x| - 1, 0\}$. Note that this regularizer is not s.p.d.*

**Corollary 2.** *Let $\dim(\mathcal{H}) \geq 2$, then the p.h. function $f \in \Gamma_0$ satisfies any item of Theorem 4 iff it is a positive multiple of the norm $\|\cdot\|$.*

*Proof:* [10, Theorem 4] showed that under positive homogeneity, i) implies that $f$ is a positive multiple of the norm. ∎

Therefore (positive multiples of) the Hilbertian norm is the only p.h. convex function $f$ that satisfies $\mathsf{P}_{f+\kappa} = \mathsf{P}_f \circ \mathsf{P}_\kappa$ for all gauge $\kappa$. In particular, this means that the norm $\|\cdot\|$ is s.p.d. Moreover, we easily recover the following result that is perhaps not so obvious at first glance:

**Corollary 3** (Jenatton et al. [7])**.** *Fix the orthonormal basis $\{\mathbf{e}_i\}_{i \in I}$ of $\mathcal{H}$. Let $\mathcal{G} \subseteq 2^I$ be a collection of tree-structured groups, that is, either $\mathsf{g} \subseteq \mathsf{g}'$ or $\mathsf{g}' \subseteq \mathsf{g}$ or $\mathsf{g} \cap \mathsf{g}' = \emptyset$ for all $\mathsf{g}, \mathsf{g}' \in \mathcal{G}$. Then*

$$\mathsf{P}_{\sum_{i=1}^m \|\cdot\|_{\mathsf{g}_i}} = \mathsf{P}_{\|\cdot\|_{\mathsf{g}_1}} \circ \cdots \circ \mathsf{P}_{\|\cdot\|_{\mathsf{g}_m}},$$

*where we arrange the groups so that $\mathsf{g}_i \subset \mathsf{g}_j \implies i > j$, and the notation $\|\cdot\|_{\mathsf{g}_i}$ denotes the Hilbertian norm that is restricted to the coordinates indexed by the group $\mathsf{g}_i$.*

*Proof:* Let $f = \|\cdot\|_{\mathsf{g}_1}$ and $\kappa = \sum_{i=2}^m \|\cdot\|_{\mathsf{g}_i}$. Clearly they are both p.h. (and convex). By the tree-structured assumption we can partition $\kappa = \kappa_1 + \kappa_2$, where $\mathsf{g}_i \subset \mathsf{g}_1$ for all $\mathsf{g}_i$ appearing in $\kappa_1$ while $\mathsf{g}_j \cap \mathsf{g}_1 = \emptyset$ for all $\mathsf{g}_j$ appearing in $\kappa_2$. Restricting to the subspace spanned by the variables in $\mathsf{g}_1$ we can treat $f$ as the Hilbertian norm. Apply Theorem 4 we obtain $\mathsf{P}_{f+\kappa_1} = \mathsf{P}_f \circ \mathsf{P}_{\kappa_1}$. On the other hand, due to the non-overlapping property, nothing will be affected by adding $\kappa_2$, thus

$$\mathsf{P}_{\sum_{i=1}^m \|\cdot\|_{\mathsf{g}_i}} = \mathsf{P}_{\|\cdot\|_{\mathsf{g}_1}} \circ \mathsf{P}_{\sum_{i=2}^m \|\cdot\|_{\mathsf{g}_i}}.$$

We can clearly iterate the argument to unravel the proximal map as claimed. ∎

For notational clarity, we have chosen not to incorporate weights in the sum of group seminorms: Such can be absorbed into the seminorm and the corollary clearly remains intact. Our proof also reveals the fundamental reason why Corollary 3 is true: The $\ell_2$ norm admits the decomposition (5) for any gauge $g$! This fact, to the best of our knowledge, has not been recognized previously.

### 3.4 Cone Invariance

In the previous subsection, we restricted the subdifferential of $g$ to be constant along each ray. We now generalize this to cones. Specifically, consider the gauge function

$$\kappa(\mathbf{x}) = \max_{j \in J} \langle \mathbf{a}_j, \mathbf{x} \rangle, \tag{13}$$

where $J$ is a finite index set and each $\mathbf{a}_j \in \mathcal{H}$. Such polyhedral gauge functions have become extremely important due to the work of Chandrasekaran et al. [19]. Define the polyhedral cones

$$K_j = \{\mathbf{x} \in \mathcal{H} : \langle \mathbf{a}_j, \mathbf{x} \rangle = \kappa(\mathbf{x})\}. \tag{14}$$

Assume $K_j \neq \emptyset$ for each $j$ (otherwise delete $j$ from $J$). Since $\partial \kappa(\mathbf{x}) = \{\mathbf{a}_j | j \in J, \mathbf{x} \in K_j\}$, the sufficient condition (10) becomes

$$\forall j \in J, \ \mathsf{P}_f(K_j) \subseteq K_j. \tag{15}$$

In other words, each cone $K_j$ is "fixed" under the proximal map of $f$. Although it would be very interesting to completely characterize $f$ under (15), we show that in its current form, (15) already implies many known results, with some new generalizations falling out naturally.

**Corollary 4.** *Denote $E$ a collection of pairs $(m, n)$, and define the total variational norm $\|\mathbf{x}\|_{\text{tv}} = \sum_{\{m,n\} \in E} w_{m,n} |x_m - x_n|$, where $w_{m,n} \geq 0$. Then for any permutation invariant function[4] $f$,*

$$\mathsf{P}_{f+\|\cdot\|_{\text{tv}}} = \mathsf{P}_f \circ \mathsf{P}_{\|\cdot\|_{\text{tv}}}.$$

*Proof:* Pick an arbitrary pair $(m, n) \in E$ and let $\kappa = |x_m - x_n|$. Clearly $J = \{1, 2\}, K_1 = \{x_m \geq x_n\}$ and $K_2 = \{x_m \leq x_n\}$. Since $f$ is permutation invariant, its proximal map $\mathsf{P}_f(\mathbf{x})$ maintains the order of $\mathbf{x}$, hence we establish (15). Finally apply Proposition 2 and Theorem 1. ∎

**Remark 3.** *The special case where $E = \{(1, 2), (2, 3), \ldots\}$ is a chain, $w_{m,n} \equiv 1$ and $f$ is the $\ell_1$ norm, appeared first in [6] and is generally known as the fused lasso. The case where $f$ is the $\ell_p$ norm appeared in [20].*

We call the permutation invariant function $f$ symmetric if $\forall \mathbf{x}, f(|\mathbf{x}|) = f(\mathbf{x})$, where $|\cdot|$ denotes the componentwise absolute value. The proof for the next corollary is almost the same as that of Corollary 4, except that we also use the fact $\text{sign}([\mathsf{P}_f(\mathbf{x})]_m) = \text{sign}(x_m)$ for symmetric functions.

**Corollary 5.** *As in Corollary 4, define the norm $\|\mathbf{x}\|_{\text{oct}} = \sum_{\{m,n\} \in E} w_{m,n} \max\{|x_m|, |x_n|\}$. Then for any symmetric function $f$, $\mathsf{P}_{f+\|\cdot\|_{\text{oct}}} = \mathsf{P}_f \circ \mathsf{P}_{\|\cdot\|_{\text{oct}}}$.*

**Remark 4.** *This norm $\| \cdot \|_{\text{oct}}$ is proposed in [21] for feature grouping. Surprisingly, Corollary 5 appears to be new. The proximal map $\mathsf{P}_{\| \cdot \|_{\text{oct}}}$ is derived in [22], which turns out to be another decomposition result. Indeed, for $i \geq 2$, define $\kappa_i(\mathbf{x}) = \sum_{j \leq i-1} \max\{|x_i|, |x_j|\}$. Thus*

$$\| \cdot \|_{\text{oct}} = \sum_{i \geq 2} \kappa_i.$$

*Importantly, we observe that $\kappa_i$ is symmetric on the first $i-1$ coordinates. We claim that*

$$\mathsf{P}_{\| \cdot \|_{\text{oct}}} = \mathsf{P}_{\kappa_{|I|}} \circ \ldots \circ \mathsf{P}_{\kappa_2}.$$

*The proof is by recursion: Write $\| \cdot \|_{\text{oct}} = f + g$, where $f = \kappa_{|I|}$. Note that the subdifferential of $g$ depends only on the ordering and sign of the first $|I| - 1$ coordinates while the proximal map of $f$ preserves the ordering and sign of the first $|I| - 1$ coordinates (due to symmetry). If we pre-sort $\mathbf{x}$, the individual proximal maps $\mathsf{P}_{\kappa_i}(\mathbf{x})$ become easy to compute sequentially and we recover the algorithm in [22] with some bookkeeping.*

**Corollary 6.** *As in Corollary 3, let $\mathcal{G} \subseteq 2^I$ be a collection of tree-structured groups, then*

$$\mathsf{P}_{\sum_{i=1}^m \| \cdot \|_{g_i,k}} = \mathsf{P}_{\| \cdot \|_{g_1,k}} \circ \cdots \circ \mathsf{P}_{\| \cdot \|_{g_m,k}},$$

*where we arrange the groups so that $g_i \subset g_j \implies i > j$, and $\|\mathbf{x}\|_{g_i,k} = \sum_{j=1}^k |x_{g_i}|_{[j]}$ is the sum of the $k$ (absolute-value) largest elements in the group $g_i$, i.e., Ky-Fan's $k$-norm.*

*Proof:* Similar as in the proof of Corollary 3, we need only prove that

$$\mathsf{P}_{\| \cdot \|_{g_1,k} + \| \cdot \|_{g_2,k}} = \mathsf{P}_{\| \cdot \|_{g_1,k}} \circ \mathsf{P}_{\| \cdot \|_{g_2,k}},$$

where w.l.o.g. we assume $g_1$ contains all variables while $g_2 \subset g_1$. Therefore $\| \cdot \|_{g_1,k}$ can be treated as symmetric and the rest follows the proof of Corollary 5. ∎

Note that the case $k \in \{1, |I|\}$ was proved in [7] and Corollary 6 can be seen as an interpolation. Interestingly, there is another interpolated result whose proof should be apparent now.

**Corollary 7.** *Corollary 6 remains true if we replace Ky-Fan's $k$-norm with*

$$\|\mathbf{x}\|_{\text{oct},k} = \sum_{1 \leq i_1 < i_2 < \ldots < i_k \leq |I|} \max\{|x_{i_1}|, \ldots, |x_{i_k}|\}. \tag{16}$$

Therefore we can employ the norm $\|\mathbf{x}\|_{\text{oct},2}$ for feature grouping in a hierarchical manner. Clearly we can also combine Corollary 6 and Corollary 7.

**Corollary 8.** *For any symmetric $f$, $\mathsf{P}_{f + \| \cdot \|_{\text{oct},k}} = \mathsf{P}_f \circ \mathsf{P}_{\| \cdot \|_{\text{oct},k}}$. Similarly for Ky-Fan's $k$-norm.*

**Remark 5.** *The above corollary implies that Ky-Fan's $k$-norm and the norm $\| \cdot \|_{\text{oct},k}$ defined in (16) are both s.p.d. (see Definition 1). The special case for the $\ell_p$ norm where $p \in \{1, 2, \infty\}$ was proved in [23, Proposition 11], with a substantially more complicated argument. As pointed out in [23], s.p.d. regularizers allow us to perform lazy updates in gradient-type algorithms.*

We remark that we have not exhausted the possibility to have the decomposition (5). It is our hope to stimulate further work in understanding the prox-decomposition (5).

**Added after acceptance:** We have managed to extend the results in this subsection to the Lovász extension of submodular set functions. Details will be given elsewhere.

## 4   Conclusion

The main goal of this paper is to understand when the proximal map of the sum of functions decomposes into the composition of the proximal maps of the individual functions. Using a simple sufficient condition we are able to completely characterize the decomposition when certain scaling invariance is exhibited. The generalization to cone invariance is also considered and we recover many known decomposition results, with some new ones obtained almost effortlessly. In the future we plan to generalize some of the results here to nonconvex functions.

## Acknowledgement

The author thanks Bob Williamson and Xinhua Zhang from NICTA—Canberra for their hospitality during the author's visit when part of this work was performed; Warren Hare, Yves Lucet, and Heinz Bauschke from UBC—Okanagan for some discussions around Theorem 4; and the reviewers for their valuable comments.

## Footnotes

[1]In some sense, this procedure is to compute $\mathsf{P}_{f+g} \approx \lim_{t \to \infty}(\mathsf{P}_f \circ \mathsf{P}_g)^t$, modulo some intermediate steps. Essentially, our goal is to establish the one-step convergence of that iterative procedure.

[2] A gauge is a positively homogeneous convex function that vanishes at the origin.

[3]Note that $\lambda \leq 1$ is necessary since any proximal map is nonexpansive.

[4]All we need is the weaker condition: For all $\{m, n\} \in E$, $x_m \geq x_n \implies [\mathsf{P}_f(\mathbf{x})]_m \geq [\mathsf{P}_f(\mathbf{x})]_n$.

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
