[Reviews · NeurIPS 2013]

Submitted by Assigned_Reviewer_4

The paper deals with an interesting theoretical question concerning the proximity operator. It investigates when the proximity of the sum of two convex functions decomposes into the composition of the corresponding proximity operators. The problem is interesting since in the applications there is a growing interest in building complex regularizers by adding several simple terms.
They pursues a quite complete study. After proving a simple sufficient condition (Theorem 1), they gives the main result of the paper (Theorem 4): it is a complete characterization of the property (for a function) of being radial versus the property of being ``well-coupled'' with positively homogeneous functions (where well-coupled means that the prox of the sum of the couple decomposes into the composition of the two individual prox map). They also consider the case of polyhedral gauge functions, deriving a sufficient condition which is expressed by means of a cone invariance property. Examples are provided which show several prox-decomposition results, recovering known facts (in a simpler way) but also proving new ones.

The value of the paper is mainly on the theoretical side. It sheds light on the mechanism of composing proximity operators and unifies several particular results that were spread in the literature. The article is well written and technically sound. The only fault I see is that perhaps some times is not completely rigorous as I explain in the following.

TYPOS, SMALL CORRECTIONS AND SUGGESTIONS:

-- Before formula (4), I would add the sentence that the Moreau envelope $M_f$ is Frechet differentiable and $\nabla M_f = P_{f^*}$.

-- after formula (5), replace ``$f \circ g$ denotes the function composition'' with ``$P_f \circ P_g$ denotes the mapping composition''.

-- For the formula $P_{f+g} = (P_{2 f}^{-1} + P_{2 g}^{-1})^{-1} \circ 2 Id$ to be valid, one needs to guarantee in some way that $\partial (f + g) = \partial f + \partial g$ --- for instance assuming 0 $\in sri(dom f - dom g)$.

-- The proof of Proposition 1, might be more direct if one relies on Proposition 2.4 in [13].

-- The equation at the end of Example 1, expreses actually the firmly non-expansivity of the composition $P_f \circ P_g$, which is a property satisfied by any proximity operators. This should be mentioned somewhere. I find the reference to [11, Eq. (5.3)] a little bit vague. For completeness one can refer more explicitly to Def 2.3 and Lemma 2.4 in [3].

-- In section 3.1, the condition $dom f \cap dom g$ non void should be assumed. It could be stated at the beginning of the section.

-- Statement of Proposition 2. The set $K_f$ is clearly a cone. I am not sure that it is also convex. Again, note that $\partial g_1 (x) + \partial g_2(x)$ in general is only strictly included in $\partial (g_1 + g_2)(x)$. However if e.g.~one of the $g_i$ is finite on the whole space, then $g_i \in K_f \implies g_1 + g_2 \in K_f$. This is the way this proposition is used later in Corollary 4.

-- The result stated in Proposition 3 is already in Moreau[11]. Thus the proof could be omitted, just refer to the article.

-- The first equation in section 3.3 is not completely rigorous. What is $g^\prime(s x)$? the directional derivative or a subgradient? Moreover $\partial g(s x)$ is a subset, so what does it mean $\langle \partial g(s x), x \rangle$?

-- Before the sufficient condition (15), I would add that $\partial k(x) = \{ a_j \vert j \in J, x \in K_j\}$. This would help to understand why the sufficient condition in Theorem 1 becomes (15).

MORE CRITICAL POINTS:


-- Formula (6) would require that $\partial (f+ g) = \partial f + \partial g$. However the condition in Theorem 1 is sufficient even if one only assumes $dom f \cap dom g$ non void. Just replace (6) with $P_{f + g}(y) - y + \partial (f+g) (P_{f+g}(y)) \ni 0$. Then use (9), the sufficient condition, and the fact that $\partial f (P_f(P_g(y))) + \partial g(P_f(P_g(y))) \subset \partial (f + g) (P_f (P_g(y)))$.

-- I cannot see the reason for the emphasis put on Proposition 4 and the need to derive it by means of the (not so simple) ``trick'' discussed in Remark 3. Indeed Proposition 4 is a quite trivial result that follows directly from the changing formula for prox (see e.g.~Proximal Spitting Methods in Signal Processing by Combettes and Pesquet) and Proposition 3. Note also that the hypothesis $0 \in dom f$ should be added.
Summary: The work provides a complete closed and original study on the problem of prox-decomposition and, due to the relevance of the proximal methods in the NIPS community, I recommend this article for acceptance.

Submitted by Assigned_Reviewer_5

This authors present conditions on when the proximal map of a sum of functions decomposes into the composition of the proximal maps of the individual functions. A few known results as well as several new decompositions are their special cases.

Quality: The paper is technically sound.

Clarity: The paper is well-organized.

Originality: The conditions on decomposing the proximal map is interesting and important.

Significance: The proximal map is one of the most important components in many first-order methods. A cheap closed-form solution is essential for the success of these methods. This paper provides a few theoretical results on understanding the proximal map.
Summary: The conditions on decomposing the proximal map is interesting and important.

Submitted by Assigned_Reviewer_6

This paper establishes necessary and sufficient conditions for a proximal map
of a sum of regularizers be the decomposable as the composition of each
regularizer's proximal map.

Overall, this is a very nice paper that unifies several previous results
regarding decomposition of proximal maps, and also discovers new decompositions
arising from the established conditions. The paper is clearly written and
everything is defined and stated rigorously. It is likely that this paper becomes
influential in the development and characterization of proximal methods for
learning.

It might be worth pointing out that decomposition of the proximal map is not
always crucial for proximal gradient algorithms. E.g., online prox-grad algorithms
(such as Duchi and Singer [20]) are still fine if we compose the proximal maps,
even when the decomposition does not hold. See Lemma 2 in

A. Martins, N. Smith, E. Xing, P. Aguiar, and M. Figueiredo.
"Online Learning of Structured Predictors with Multiple Kernels."
AISTATS 2011.

(Some additional results are in the first author's thesis, sect. 9.3.4.)

Minor comments:
- Prop. 3 could be made clearer. X iff Y iff Z. I believe this means that X, Y, Z are
all equivalent, but it's not immediately clear if the intended meaning is
instead (X iff Y) iff Z or even X iff (Y iff Z).
- line 205: \lambda \in [0,1]^2 -> bad place to put a footnote! (see http://xkcd.com/1184/)
- in theo. 4, should probably use \lambda(u) instead of \lambda to emphasize
dependency on u
- I found remark 3 very cryptic... Maybe the argument can be clarified?
- Prop. 4: doesn't (ii) imply (i) and (iii)? Maybe you should just state (ii) and then mention (i) and (iii) as particular cases
- example 5: "gauge" is never defined rigorously. Do you mean positive homogeneous?
- in Corollary 2, it would be useful to add subscript ||.||_{\mathcal{H}} to emphasize this is the Hilbertian norm
Summary: This paper establishes necessary and sufficient conditions for a proximal map
of a sum of regularizers be the decomposable as the composition of each
regularizer's proximal map.

Overall, this is a very nice paper that unifies several previous results
regarding decomposition of proximal maps, and also discovers new decompositions
arising from the established conditions. The paper is clearly written and
everything is defined and stated rigorously. It is likely that this paper becomes
influential in the development and characterization of proximal methods for
learning.
Author Feedback

Author rebuttal: We thank all reviewers for their valuable comments, and we address some of the concerns below.

=================================================
Assigned_Reviewer_4:

Q1: the additive property of subdifferential needs assumption.

A: Very true. We omitted this point mainly because of the space limit and its very pathological nature. In most cases (at least) one of the regularizer f or g is continuous hence the additive property holds. We will add a comment on this.


Q2: shorten the proof of Prop. 1

A: In fact, the first proof we had was exactly what the reviewer suggested. The reason to use a slightly different proof is:
1) we did not want to introduce firmly nonexpansiveness since it is less familiar to many NIPS audience;
2) Moreau's characterization of the proximal map, in our opinion, is very useful but under-appreciated. Somehow we feel obligated to use this result and popularize it. Of course, this is personal taste and certainly debatable.


Q3: proof of Prop. 3 can be omitted

A: Correct, this result is due to Moreau (we had to put the reference in line 161 due to some Latex issue). The intention to include a brief proof is for the convenience of non-French readers (as the only source we found about the proof is Moreau's original paper).


Q4: g' and the meaning of the integration

A: g, as a function of the scalar t, is 1-D, hence in integration it does not matter how we interpret g' (be subgradient or directional derivative). The subdifferential in the integration is meant to be an arbitrary subgradient selector (of course, to be completely rigorous we need a measurable selector). Since this paragraph is meant to motivate the consideration of scaling invariance and positive homogeneity, we prefer not to make the argument painfully rigorous (although can be done thanks to convexity). Will add some clarification.


Q5: slight improvement of the sufficient condition

A: Very good point. Will incorporate it.


Q6: Prop. 4 is trivial

A: Very good point. Will remove it and save some space.


Other comments will be taken into account in the revision.



=================================================
Assigned_Reviewer_5:

Thanks for the positive feedback.



=================================================
Assigned_Reviewer_6:

Q1: Prop. 4 can be shortened

A: ii) implies i) and iii) in terms of sufficiency but not necessity. As pointed out by Assigned_Reviewer_4, our proof is unnecessarily complicated. Will remove this part.


Q2: Remark 3 is vague

A: The main idea is this: We want to prove object A enjoys some property P. So first reduce object A to object B which we easily prove to possess P. If the inverse map (from B to A) preserves P, we immediately know A also possesses P. Due to the removal of Prop. 4, we will revise this remark accordingly.

Q3: definition of gauge

A: Correct, gauge means positive homogeneous convex (vanishing at 0). Will mention explicitly.


Q4: Lemma 2 in Martins et. al

A: Very interesting reference. A quick look gives us the impression that the reason why the composition (even when it does not hold) always works is because of the slower rate of subgradient algorithms. We will add this reference and take some time in understanding it more thoroughly.


Other comments will be taken into account in the revision.